# The Superficial Peroneal Nerve Is at Risk during the “All Inside” Arthroscopic Broström Procedure: A Cadaveric Study

**DOI:** 10.3390/medicina59061109

**Published:** 2023-06-08

**Authors:** Sung Hwan Kim, Jae Hyuck Choi, Sang Heon Lee, Young Koo Lee

**Affiliations:** 1Department of Orthopaedic Surgery, Soonchunhyang University Hospital Bucheon, 170, Jomaru-ro, Wonmi-gu, Bucheon-si 14584, Republic of Korea; shk9528@naver.com (S.H.K.); worldking70@naver.com (S.H.L.); 2Department of Orthopedics, Manjok Clinic, 178, Jibeom-ro, Suseong-gu, Daegu 42208, Republic of Korea; manjokclinic@naver.com

**Keywords:** arthroscopic modified Broström procedure, ankle anterolateral portal, cadaver, intermediate superficial peroneal nerve

## Abstract

*Background*: The arthroscopic Broström procedure is a promising treatment for chronic ankle instability. However, little is known regarding the location of the intermediate superficial peroneal nerve at the level of the inferior extensor retinaculum; knowledge about this location is important for procedural safety. The purpose of this cadaveric study was to clarify the anatomical relationship between the intermediate superficial peroneal nerve and the sural nerve at the level of the inferior extensor retinaculum. *Methods*: Eleven dissections of cadaveric lower extremities were performed. The origin of the experimental three-dimensional axis was defined as the location of the anterolateral portal during ankle arthroscopy. The distances from the standard anterolateral portal to the inferior extensor retinaculum, sural nerve, and intermediate superficial peroneal nerve were measured using an electronic digital caliper. The location of inferior extensor retinaculum, the tract of sural nerve, and intermediate superficial peroneal nerve were checked using average and standard deviations. For the statistical analyses, data are presented as average ± standard deviation, and then they are reported as means and standard deviations. Fisher’s exact test was used to identify statistically significant differences. *Results*: At the level of the inferior extensor retinaculum, the mean distances from the anterolateral portal to the proximal and distal intermediate superficial peroneal nerve were 15.9 ± 4.1 (range, 11.3–23.0) mm and 30.1 ± 5.5 (range, 20.8–37.9) mm, respectively. The mean distances from the anterolateral portal to the proximal and distal sural nerve were 47.6 ± 5.7 (range, 37.4–57.2) mm and 47.2 ± 4.1 (range, 41.0–51.8) mm), respectively. *Conclusions*: During the arthroscopic Broström procedure, the intermediate superficial peroneal nerve may be damaged by the anterolateral portal; the proximal and distal parts of the intermediate superficial peroneal nerve were located within 15.9 and 30.1 mm, respectively, at the level of the inferior extensor retinaculum in cadavers. These areas should be considered danger zones during the arthroscopic Broström procedure.

## 1. Introduction

Ankle sprains, particularly lateral inversion sprains, are the most common ankle injuries; they constitute 85% of all cases [1]. The lateral ligament complex is typically involved in these injuries, which can vary from a microscopic tear to a complete tear [2]. In cases of inversion sprain, the most commonly injured ligament is the anterior talofibular ligament (ATFL), which stabilizes the ankle in plantar flexion during inversion stress. Acute injuries typically respond to conservative treatment, including immobilization and functional rehabilitation. However, up to 34% of patients experience repeated ankle sprains, leading to chronic ankle instability in 5–20% of cases [3,4]. The presence of pain and subjective sense of instability may result in additional ankle injuries. Chronic ankle instability may alter the ankle joint biomechanics, leading to cartilage degeneration and secondary osteoarthritis, which require surgical intervention [5]. In particular, varus ankle instability caused by medially shifting contact pressure can cause osteochondral lesions of the talus in the medial talar dome. The lateral ligament complex of the ankle consists of the ATFL, calcaneofibular ligament (CFL), and posterior talofibular ligament (PTFL). The ATFL and CFL are mainly responsible for ankle instability. The ATFL originates at the inferior segment of the anterior border of the distal fibula and inserts on the body of the talus, anterior to the lateral articular surface. The CFL originates immediately distal to the inferior band of the ATFL and inserts on the lateral calcaneal surface [6]. The ATFL prevents inversion and anterior displacement of the talus and the CFL stabilizes the subtalar joint. The lateral ligament complex is most commonly injured by inversion and plantar flexion. Although isolated ATFL ruptures are common, moderate to severe injuries may involve both the ATFL and CFL. Isolated CFL ruptures are uncommon and have rarely been reported [7].

There have been many studies and reports on effective surgical methodologies for correction of chronic ankle instability, including the Broström technique, with or without Gould modification, as well as the Watson-Jones, Evans, Larsen, Chrisman-Snook, and Pisani. Open surgery and arthroscopy are associated with good outcomes. Recently, there has been increasing implementation of minimally invasive surgery using arthroscopy, including anatomical repair and reconstruction by means of arthroscopy and percutaneous or mini-open techniques that do not require the use of an arthroscope. In 1966, Broström described an anatomical surgical procedure, which directly repairs the injured ligament in relation to using ATFL remnants to avoid muscle imbalance. In 1980, Gould modified this procedure by reinforcing the inferior extensor retinaculum to improve mechanical strength and to overcome the limitations of previous procedures, including procedural complexity, prolonged immobilization requirement, and ankle degeneration risk. The main advantages of the Gould modification are its simplicity, ability to preserve subtalar joint mobility and restore the physiological joint anatomy and kinematics, and association with fewer intraoperative and postoperative complications relative to conventional techniques [8]. Additionally, the Gould modification does not require loss of other tendons, thus preventing ankle degeneration related to non-anatomical forces. Currently, the Gould modification of the Broström procedure is regarded as the standard treatment for chronic ankle instability [9,10]. This procedure entails augmented Broström repair with additional reinforcements of the inferior extensor retinaculum, calcaneofibular ligament, and the lateral talocalcaneal ligament.

Arthroscopic ligament repair is increasingly utilized because it allows assessments of intra-articular pathology; it also leads to rapid recovery and low morbidity. The arthroscopic Broström procedure involves reinforcing the inferior extensor retinaculum with an anchor suture through the insertion of conventional anteromedial and anterolateral portals for diagnostic arthroscopy, as well as an accessory anterolateral portal that is inserted near the fibula tip. The suture anchor and instruments are passed through the anterolateral portal for ligament repair. This portal is inserted lateral to the peroneus tertius or the extensor digitorum longus tendon at the anterior joint line. Although this technique is associated with acceptable outcomes and is minimally invasive, complications may occur in 14–17% of cases [11]. Frekel published a review of complications of foot and ankle arthroscopy in 2001 [12]. They reported 612 cases of ankle arthroscopic procedures and found that complications occurred in 9% of patients undergoing ankle arthroscopy; half of the complications were neurological injuries, and 27% were iatrogenic superficial peroneal nerve injuries. Neurovascular complications are more common after ankle arthroscopy than after arthroscopy, involving other joints, such as the knees and shoulders. Ankle arthroscopic surgery can cause not only nerve damage, but also vascular damage. As the branches of the intermediate superficial peroneal nerve reported the most commonly damaged nerve, anterior tibial arteries and their branches showed their vulnerability in ankle arthroscopic surgery. Pseudoaneurysm of the anterior tibial artery after ankle arthroscopic surgery was reported by several studies [13,14]. Son et al. [14] reported anatomic analysis of the anterior tibial artery by using magnetic resonance imaging. In 6.2% of the 258 cases, the anterior tibial artery and its branches were located near the anterolateral portal, which introduces the risk of vascular damage during ankle arthroscopic surgery. In addition, the mean distance between the anterior tibial artery and the joint capsule was only 2.3 mm, which was very close to the anterior working space of the ankle joint. Ankle plantarflexion during portal placement significantly increases the distance between the malleolar artery and portal, thus diminishing the potential for injuries to the malleolar arteries [15].

Branches of the intermediate superficial peroneal nerve or sural nerve may be damaged by the anterolateral portal [16,17]. The superficial peroneal nerve provides motor innervation to the peroneus longus and brevis muscles. It passes through the lateral intermuscular septum, pierces the crural fascia, and provides the sensory supply to most of the dorsal foot. After piercing the fascia, it divides into two terminal branches: the medial dorsal cutaneous nerve and the intermediate dorsal cutaneous nerve [18]. Anatomic variations of the superficial peroneal nerve make it easier for neurological complications to occur in ankle arthroscopic surgery. Prakash et al. [19] reported a cadaver study associated with anatomic variations, including the course and the distribution of the superficial peroneal nerve. As we often know, the superficial peroneal nerve is located in the lateral compartment of the leg. However, according to their study, the superficial peroneal nerve was located in the anterior compartment of the leg in 28% of specimens. Additionally, in 20% of specimens, the superficial peroneal nerve did not divide into terminal branches after it had pierced the deep fascia. The superficial peroneal nerve branched before piercing the peroneus longus and extensor digitorum longus muscles, or it branched after piercing these two muscles and before piercing the deep fascia. In 33% of specimens, there was an additional branch from the sensory division of superficial peroneal nerve, which may be called the accessory deep peroneal nerve. It courses in the anterior compartment of the leg and supplies the ankle and the dorsum of foot. Other studies have reported similar proportions of anatomical variations of superficial peroneal nerve to this study [20,21,22].

There is a need to identify the locations of the inferior extensor retinaculum, intermediate superficial peroneal nerve, and sural nerve to prevent damage to these structures. The “nick and spread” technique is commonly used to prevent nerve damage by arthroscopic portals. However, the nerves are easily injured at the level of the inferior extensor retinaculum during ankle arthroscopy [23]. Previous studies have shown that the most common complication of the arthroscopic Broström procedure is damage to the intermediate superficial peroneal nerve [24]. Multiple studies have explored the course and distribution of the superficial peroneal nerve [25,26]. However, little is known regarding the location of the intermediate superficial peroneal nerve at the level of the inferior extensor retinaculum, where the intermediate superficial peroneal nerve may be injured during the arthroscopic Broström procedure. Although nerve damage after ankle arthroscopy is uncommon, such damage leads to severe patient discomfort and may be irreversible in severe cases. Additionally, it may negate the advantages of minimally invasive arthroscopic surgery. As a result, neural complications of ankle arthroscopy should not be overlooked. Masato et al. [27] reported a case of superficial peroneal nerve injury during ankle arthroscopy in a 20-year-old woman. She developed pain in the dorsum of the foot, which radiated from the insertion site of the anterolateral portal to the dorsomedial aspect of the foot. Because conservative treatment was ineffective, the patient underwent additional surgery, which revealed a neuroma in the intermediate superficial peroneal nerve.

It is important to develop strategies that can prevent nerve damage during the arthroscopic Broström procedure. Here, we evaluated the anatomical relationships of anterolateral portal sites with the intermediate superficial peroneal nerve and the sural nerve at the level of the inferior extensor retinaculum to identify the safest portal insertion sites for the arthroscopic Broström procedure.

## 2. Materials and Methods

This study was conducted using human cadavers. Therefore, research ethics review, such as IRB approval or the consent process of the subjects, were not conducted. In total, 11 embalmed frozen foot specimens without visible deformity or pathology were obtained from four male and two female cadavers with a mean foot size of 230.2 (range, 225–250) mm. The ankles were fixed at 90° with fixing plates. No limbs showed evidence of previous surgeries. The anterolateral portal site was immediately lateral to the peroneus tertius at the level of the ankle joint. After removal of the skin, the ankle was sequentially dissected; the distances from anterolateral portal point to the inferior extensor retinaculum, intermediate superficial peroneal nerve, and sural nerve were measured using an electronic digital caliper (CD-15CP; Mitutoyo Corp., Tokyo, Japan). Measurements (mm) were obtained to the nearest 0.1 mm.

Kirschner wires (K-wires) with an electric drill were used to mark each point. For the pointing to standard anterolateral portal, a K-wire was fixed directly anteriorly to posteriorly as *x*-axis. The *x*-axis runs anteriorly to posteriorly, the *y*-axis runs laterally to medially, and the *z*-axis runs runs superiorly to inferiorly. Each angle, verified by goniometry, is exactly 90-degrees (Figure 1).

The inferior extensor retinaculum was located along the labeled *x*-axis at the vertical right angle. The proximal and distal parts of the inferior extensor retinaculum were labeled as A1 and A2, respectively. The distances between anterolateral portal sites and A1 and A2 were measured. The sural nerve passes through the inferior extensor retinaculum, curves around the lateral malleolus, and divides into medial and lateral branches at the base of the fifth metatarsal. The proximal and distal contact points of the sural nerve and inferior extensor retinaculum are designated B1 and B2, respectively. The A1, A2, B1, and B2 points were joined to form an imaginary quadrangle on the inferior extensor retinaculum edge. The proximal and distal points of intersection of intermediate superficial peroneal nerve and inferior extensor retinaculum were designated D1 and D2, respectively.

We measured the distances between the nearest inferior extensor retinaculum edges in front of A1 and the anterolateral portal (O), between the farthest inferior extensor retinaculum edges in front of A2 and O, between the nearest intermediate superficial peroneal nerve branch locations at D1 and O, between the farthest intermediate superficial peroneal nerve locations at D2 and O, between the nearest sural nerve locations at B1 and O; between the farthest sural nerve locations at B2 and O, and between A1–B1, B1–B2, A2–B2, A1–A2, A1–D1, A2–D2, B1–D1, and B2–D2. The distances from the anterolateral portal to various points were measured using an electronic digital caliper.

A1 (blue), A2 (purple), B1 (yellow), B2 (brown), D1 (aqua), and D2 (red) are presented as mean ± standard deviation (range, min–max). Maya software was used for 3D rendering and mapping of our results (Figure 2).

Data were analyzed using Excel (Microsoft Corp., Redmond, WA, USA). Data are presented as means ± standard deviations. Fisher’s exact test was used to identify statistically significant differences. To estimate the risk to each nerve from the anterolateral portal, we assumed that the nerves would be located within two standard deviations of the mean distance from anterolateral portal in 95% of cases; this was regarded as the minimum safe distance.

## 3. Results

The mean O–A1, O–A2, D1–O, D2–O, B1–O, B2–O, A1–B1, B1–B2, A2–B2, A1–A2, A1–D1, A2–D2, B1–D1, and B2–D2 distances were 15.9 ± 11 (range, 2.6–42.6) mm, 30.1 ± 9.8 (range, 16.8–46.4) mm, 15.9 ± 4.1 (range, 11.3–23.0) mm, 30.1 ± 5.5 (range, 20.8–37.9) mm, 47.6 ± 5.7 (range, 37.4–57.2) mm, 47.2 ± 4.1 (range, 41.0–51.8) mm, 52.1 ± 10.8 (range, 40.2–76.5) mm, 28 ± 12.5 (range, 11.2–55.9) mm, 41.6 ± 10.6 (range, 24–58.9) mm, 21.5 ± 10.6 (range, 9.8–42.7) mm, 13.2 ± 4.1 (range, 0–31.2) mm, 12.3 ± 10.8 (range, 0–30.6) mm, 40.3 ± 14 (range, 15.6–60.1) mm, and 32.1 ± 13.8 (range, 17.7–61.4) mm, respectively (Table 1).

## 4. Discussion

Care is needed to avoid nerve damage, particularly to the intermediate superficial peroneal nerve, during the insertion of an accessory anterolateral portal for the modified arthroscopic Broström procedure. Our results suggest that nerve damage can be prevented by avoiding an area of 15.9 mm around the anterolateral portal.

Surgical treatment of chronic ankle instability, first described in 1932, originally involved suturing the ruptured ligament and fortifying it by means of the peroneus brevis tendon [28]. Surgical treatment options have advanced over the past 30 years; they currently include direct anatomical repair, anatomical reconstruction, and non-anatomical reconstruction. The non-anatomical procedures include the Watson-Jones and Chrisman-Snook procedures, which involve the peroneus brevis tenodesis. These procedures are usually associated with good short-term outcomes; however, some patients may experience persistent pain, stiffness, wound complications, and impaired ankle and subtalar joint function [29,30]. Therefore, the use of non-anatomical reconstruction has decreased over time. The first-line surgical treatment of chronic ankle instability is direct anatomical repair by the Broström procedure with the Gould modification. Anatomical ATFL reconstruction using an autograft or allograft can be performed in patients with generalized laxity, insufficient remnant tissue, or failed prior stabilization procedures [31].

Recent arthroscopic advancements have enabled stabilization of the lateral ankle ligament complex, repair of lateral ankle instability, and early detection of intra-articular pathology (e.g., chondral lesions and loose bodies) with minimal incisions. This minimally invasive procedure involves the placement of a suture anchor in the fibula to repair the ATFL. A cadaveric study showed no differences in strength or stiffness between conventional open repair and arthroscopic direct anatomical repair [32]. Along with the standard anteromedial and anterolateral portals, an additional lateral working portal is needed to augment the inferior extensor retinaculum during ankle arthroscopy [17,24]. Although arthroscopy has excellent outcomes, the overall complication rates are similar to or higher than the rates for open procedures [33,34]. Wang et al. [34] demonstrated that 31 of 179 patients undergoing arthroscopic ATFL repair with the suture anchor technique developed complications, including portal site irritation, delayed wound healing, and nerve complications. Nerve complications were comprised of intermediate dorsal cutaneous nerve neuritis, superficial peroneal nerve numbness, and sural nerve neuritis.

Superficial peroneal nerve injury, sural neuritis, and knot prominence can occur after arthroscopy [35]. Although wound complications are more common after open surgery, superficial peroneal nerve damage is more common after arthroscopy [36]. The high rate of sensory nerve damage during arthroscopy is explained by the presence of a communicating branch between the superficial peroneal and sural nerves inferior to the fibula [37,38]. The most common complication of arthroscopy is superficial peroneal injury, with an incidence of 3–11% [11,24]. Intermediate superficial peroneal nerve entrapment, a major complication of arthroscopic ATFL repair, is the result of insufficient knowledge regarding the course of the intermediate superficial peroneal nerve. Superficial peroneal nerve entrapment leads to pain and paresthesia over the lateral aspect of the calf and dorsum of the foot.

The superficial peroneal nerve is the only nerve in the human body that is visible from the skin surface. Combined ankle plantar flexion and inversion can cause the superficial peroneal nerve to become more prominent. Because the anterior portals for ankle arthroscopy are inserted with the ankle in the neutral or slightly dorsiflexed position, the course of the superficial peroneal nerve should be identified preoperatively by ankle plantar flexion and inversion to avoid iatrogenic nerve injury. Multiple strategies have been suggested to avoid damage to the superficial peroneal nerve [35,36]; these strategies require the operator to have good anatomical knowledge. Nerve injury may be avoided by making vertical skin incisions parallel to the tendons and nerve, performing blunt dissection up to the level of the joint, and using minimal distraction and tourniquet inflation. Prolonged distraction can lead to various complications [12]. Acevedo et al. [24] suggested a “safe zone” 1.5 cm from the fibula tip. The “nick and spread” technique can help to avoid superficial peroneal nerve damage around the portal. This technique involves the use of a hemostat to spread soft tissues down to the joint capsule. Stephens and Kelly [39] demonstrated that the superficial peroneal nerve can be identified via direct inspection and palpation near the skin surface during ankle inversion and fourth toe flexion. However, the intermediate superficial peroneal nerve was visualized in only 30% of patients during ankle plantar flexion and inversion [18]. Ucerler et al. [40] demonstrated that the nearest superficial peroneal nerve branches were located 2.2–16.2 mm from the lateral border of the peroneus tertius tendon. Although multiple studies have evaluated the complications of ankle arthroscopy, particularly nerve damage, and corresponding preventive strategies, few studies have determined the distance between the intermediate superficial peroneal nerve and the accessory anterolateral portal. Woo et al. [41] demonstrated that, at the level of the ankle joint, the distance between the anterolateral portal and the intermediate superficial peroneal nerve was short (mean distance: 5.5 [range, 0.4–14.4] mm), so it was an anatomic hazard. Charles et al. [35] found that the mean distance from the portal to the superficial peroneal nerve was 13.11 (range, 2–24) mm; they suggested that the accessory lateral working portal was a safe access point because the major at-risk structures (superficial peroneal nerve branches, sural nerve branches, and peroneal tendons) were located > 1 cm from the anterolateral portal. Our results differ from the findings in previous studies because we used intermediate superficial peroneal nerve locations nearest and farthest from the inferior extensor retinaculum for measurements. We selected the inferior extensor retinaculum because it is an important anatomical landmark during the modified arthroscopic Broström procedure. Additionally, the location of the anterolateral portal can affect the distance to the intermediate superficial peroneal nerve. We created an anterolateral portal immediately lateral to the peroneus tertius tendon, whereas Charles et al. [35] created an anterolateral portal 1.5 cm anterior to the distal tip of the fibula.

In our study, the mean distance from the anterolateral portal to the intermediate superficial peroneal nerve at the level of the inferior extensor retinaculum was 15.9 (range, 11.3–23.0) mm. The intermediate superficial peroneal nerve passes through the quadrangle space formed by the inferior extensor retinaculum. The distances from the origin to the proximal (A1) and distal (A2) inferior extensor retinaculum were 15.9 and 30.1 mm, respectively. Intermediate superficial peroneal nerve (D) was located lateral to the inferior extensor retinaculum at A1 and A2. The A1–D1 and A2–D2 mean distances were 13.2 and 12.3 mm, respectively. The A1 and A2 points were located parallel to the anterolateral portal on the *x*-axis. The danger zone was located 13.2 mm lateral to A1 and 15.9 mm from the anterolateral portal. Therefore, caution is required when approaching the inferior extensor retinaculum during the arthroscopic Broström procedure.

In the present study, we determined the anatomical relationships between the portals and the neurovascular structures surrounding the inferior extensor retinaculum. However, this study had some limitations. First, only 11 embalmed frozen foot specimens were evaluated to determine the course of the intermediate superficial peroneal nerve. Further studies with larger sample sizes are needed to generalize our results and to identify anatomical variations. Although the superficial peroneal nerve typically divides into the medial dorsal cutaneous nerve and the intermediate dorsal cutaneous nerve after it has pierced the fascia, some patients may have up to five divisions of the superficial peroneal nerve [42]. Additionally, as we mentioned earlier, there are several anatomic variations of the superficial peroneal nerve, as reported in other cadaver studies. However, we only identified the two terminal branches of the superficial peroneal nerve, and we did not consider the anatomic variations of superficial peroneal nerve, as there were no specific anatomic variations in our specimens. Second, although ankles with visible deformity, pathology, or previous surgery were excluded, concomitant conditions (e.g., soft tissue stiffness and Achilles tendon tightness) were not evaluated. Chronic ankle instability may be caused by a lack of soft tissue stiffness, which can also lead to ankle structure displacement. Third, we did not evaluate the symptoms or effects of nerve damage because this was a cadaveric dissection study. Multicenter studies with large sample sizes and the inclusion of living patients are needed to characterize the anatomical relationships of ankle arthroscopy portals with major neurovascular structures and tendons. Despite these study limitations, our results may facilitate the development of strategies to prevent iatrogenic damage to the superficial peroneal nerve and its branches during arthroscopic Broström repair. In particular, an understanding of the anatomical relationships between the inferior extensor retinaculum and the surrounding nerves can facilitate the establishment of a safe zone for the arthroscopic Broström procedure.

## 5. Conclusions

When an anchor suture is placed during the arthroscopic Broström procedure for lateral ankle instability, the inferior extensor retinaculum is located 15.9–30.1 mm from the anterolateral portal on the *x*-axis. At the level of the inferior extensor retinaculum, the shortest distance between the intermediate superficial peroneal nerve and the anterolateral portal is 15.9 mm. As a result, surgeons should place the anterolateral portal away from this danger area to prevent the injury of intermediate superficial peroneal nerve, as long as it does not interfere with surgery, and care must be taken around this danger area.

## Figures and Tables

**Figure 1 medicina-59-01109-f001:**
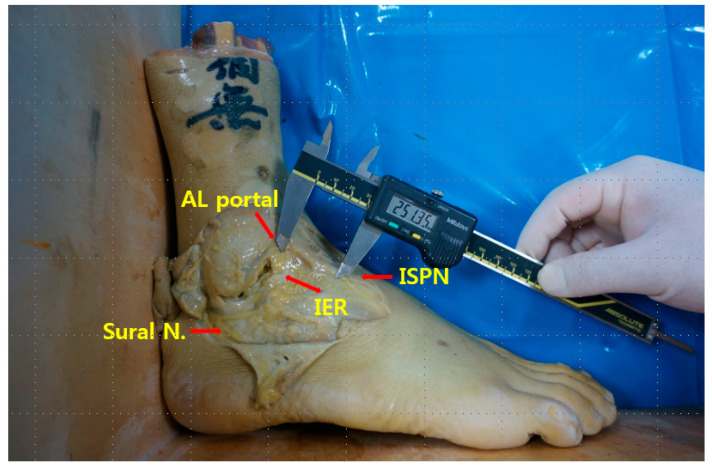
Distances from the insertion point of the anterolateral portal to the inferior extensor retinaculum), intermediate superficial peroneal nerve, and sural nerve were measured using an electronic digital caliper (CD-15CP; Mitutoyo Corp., Tokyo, Japan). That figure shows the measurement of the distance from the anterolateral portal to the farthest inferior extensor retinaculum.

**Figure 2 medicina-59-01109-f002:**
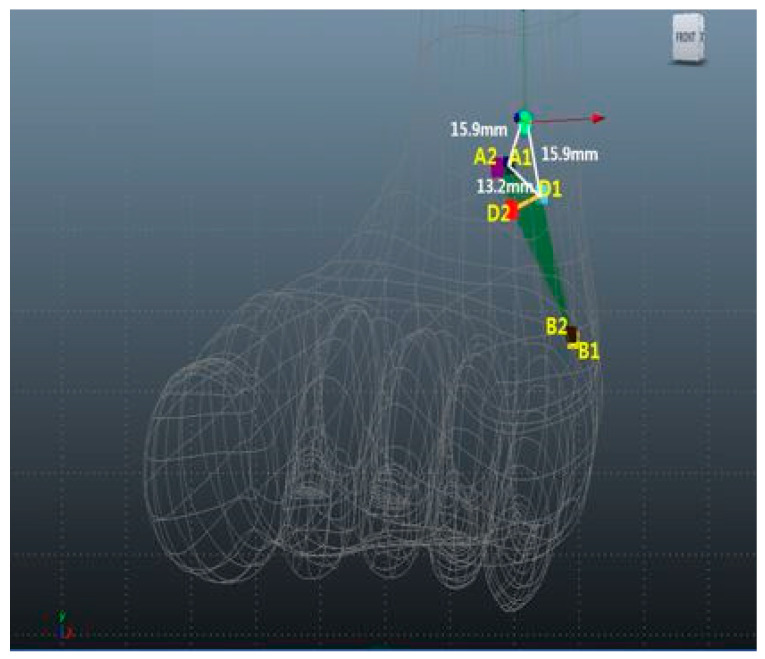
Mean distances from the anterolateral portal to *O* (AL portal sites), *A1* (blue; nearest edge of inferior extensor retinaculum), *A2* (purple; farthest inferior extensor retinaculum edge), *B1* (yellow; nearest sural nerve location on inferior extensor retinaculum), *B2* (brown; farthest sural nerve location on inferior extensor retinaculum), *D1* (aqua; nearest intermediate superficial peroneal nerve branch on inferior extensor retinaculum), and *D2* (red; farthest intermediate superficial peroneal nerve branch on inferior extensor retinaculum). The dark green quadrangle is the imaginary quadrangle, which is formed by A1, A2, B1, and B2 on the inferior extensor retinaculum.

**Table 1 medicina-59-01109-t001:** Average distances of O–A1, O–A2, D1–O, D2–O, B1–O, B2–O, A1–B1, B1–B2, A2–B2, A1–A2, A1–D1, A2–D2, B1–D1, and B2–D2. O (AL portal sites), A1 (nearest edge of inferior extensor retinaculum), A2 (farthest inferior extensor retinaculum edge), B1 (nearest sural nerve location on inferior extensor retinaculum), B2 (farthest sural nerve location on inferior extensor retinaculum), D1 (nearest intermediate superficial peroneal nerve branch on inferior extensor retinaculum), and D2 (farthest intermediate superficial peroneal nerve branch on inferior extensor retinaculum).

Point-Point	Mean Distance
O–A1	15.9 mm
O–A2	30.1 mm
D1–O	15.9 mm
D2–O	30.1 mm
B1–O	47.6 mm
B2–O	47.2 mm
A1–B1	52.1 mm
B1–B2	28.0 mm
A2–B2	41.6 mm
A1–A2	21.5 mm
A1–D1	13.2 mm
A2–D2	12.3 mm
B1–D1	40.3 mm
B2–D2	32.1 mm

## Data Availability

Data sharing is not applicable to this article because no datasets were made or analyzed during this study.

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
