# Peer review of "The Superficial Peroneal Nerve Is at Risk during the “All Inside” Arthroscopic Broström Procedure: A Cadaveric Study"

_medicina, 2023, doi:10.3390/medicina59061109_

Round 1
Reviewer 1 Report
Careful research. Good presentation of things.
Authors should add a clear drawing of the structures in Figure 1 to the article
Line 51: the authors should add a clear drawing of those ligaments
Line 115: the writers should add a clear drawing of those nerves
Authors should add an explanatory drawing with abbreviations and a table with average values without ranges in the Results section
Line 293: [11, 24] intermediate, Intermediate
Line 307: of the joint [new18] ?
Line 341: respectively. intermediate superficial Intermediate
Author Response
Thank you very much for reviewing our paper.
We think It's a really kind thing to review our paper while busy with various things.
And your review was mostly positive and complimentary words, encouraging us.
We appreciate again about your kind review and advice.
Careful research. Good presentation of things.
Authors should add a clear drawing of the structures in Figure 1 to the article
-> We agree with your comments. Based on your comments, we change Figure 1 to make the structures easier to see. Also, we add the explanation of what the following figure measures. (Line187-191)
Line 51: the authors should add a clear drawing of those ligaments
-> Thanks for your comments. Clear drawing of the lateral ligament complex may help readers to understand this research. However, there are already many great anatomical pictures in textbooks and readers can find them with a simple research. If you think adding drawing is essential, we will take the time to consider adding them.
Line 115: the writers should add a clear drawing of those nerves
-> Thanks for your comments. Understanding course of the superficial peroneal nerve and sural nerve is essential for our paper. The reference papers we have cited have already studied the course of those nerves and provide excellent drawing and clinical pictures which will allow readers to understand our research. But if you think adding drawing is necessary, we will consider adding drawing of those nerves.
Authors should add an explanatory drawing with abbreviations and a table with average values without ranges in the Results section
-> Thanks for your comments. We add a table with average values in the Results section based on your comments. (Line264-269)
Line 293: [11, 24] intermediate, Intermediate
-> Thanks for your comments. We have corrected that lower case. (Line 309)
Line 307: of the joint [new18] ?
-> We appreciate for your comments. We have corrected that sentence. (Line 323)
Line 341: respectively. intermediate superficial Intermediate
-> Thanks for your comments and we are sorry to repeat a simple mistake. We have corrected that lower case. (Line 357)
Reviewer 2 Report
I commend the authors for their research entitled "Superficial Peroneal Nerve is “AT RISK”, During the “All In-side” Arthroscopic Broström Procedure: A Cadaveric Study". In their very interesting and precisely conducted anatomic study the authors focused in measuring distances from standard antero-lateral arthroscopic portal from important neural structures during surgical treatment of injured lateral ankle ligaments. Overall the the manuscript is clearly written and the results are important for the practicing orthopedic surgeons.
Specific remarks:
Title:
"AT RISK" - Avoid upper cases and dictate when not necessary.
Introduction.
Rows 50-52: “The lateral ligament complex of the ankle consists of the anterior talofibular ligament (ATFL), calcaneofibular ligament (CFL), and posterior talofibular ligament (PTFL).”
ATFL was defined in row 42 already.
Figures.
Figure 1: Please expand the font size to be clearly visible and explain the dark green labeled imaginary quadrangle in the figure legend.
Conclusion.
This is the worst part of the manuscript and should be rewritten. Please make conclusions regarding to you results. Mention the strengths and limitations of your research. Where should the surgeon place the anterolateral portal to minimize the risk of intermediate superficial peroneal nerve injury?
Round 2
Reviewer 2 Report
The authors have improved the manuscript as suggested.